# Study on the Effects of Angiotensin Receptor/Neprilysin Inhibitors on Renal Haemodynamics in Healthy Dogs

**DOI:** 10.3390/ijms25116169

**Published:** 2024-06-03

**Authors:** Mio Ishizaka, Yurika Yamamori, Huai-Hsun Hsu, Yuichi Miyagawa, Naoyuki Takemura, Mizuki Ogawa-Yasumura

**Affiliations:** The Laboratory of Veterinary Internal Medicine II, School of Veterinary Medicine, Faculty of Veterinary Science, Nippon Veterinary and Life Science University, 1-7-1 Kyonan Cho, Musashino-shi 180-8602, Tokyo, Japan; ooooo1425360@gmail.com (M.I.); chamamoriyurika@yahoo.co.jp (Y.Y.); hsuhuaihsun@gmail.com (H.-H.H.); ymiyagawa@nvlu.ac.jp (Y.M.); nstakemura@gmail.com (N.T.)

**Keywords:** angiotensin receptor/neprilysin inhibitor, renal haemodynamics, glomerular filtration rate, renal plasma flow

## Abstract

An angiotensin receptor/neprilysin inhibitor (ARNI), a heart failure treatment, is a combination drug made up of sacubitril, a neprilysin inhibitor, and valsartan, a vascular receptor blocker. No human or veterinary studies regarding the effect of ARNI on renal haemodynamics in the absence of cardiac or renal issues exist. Therefore, we investigated the effect of ARNI on renal haemodynamics in five healthy dogs. ARNI was administered to all five dogs at an oral dose of 20 mg/kg twice daily for 4 weeks. Renal haemodynamics were assessed on the day before ARNI administration (BL), on Day 7, and on Day 28. The glomerular filtration rate (GFR) significantly increased on Day 28 compared to BL and Day 7, whereas renal plasma flow increased on Day 7 and Day 28 compared to BL. Systolic blood pressure significantly decreased between BL and Day 28. Plasma atrial natriuretic peptide (ANP) concentrations increased on Day 7 compared to BL. Additionally, ANP concentrations increased on Day 28 in three of the five dogs. Different ANP concentrations were observed in the remaining two dogs. Both urine output volume and heart rate remained relatively stable and did not exhibit significant change. In conclusion, ARNI may enhance renal haemodynamics in healthy dogs. ARNI could be a valuable drug for treating both heart and kidney disease in dogs.

## 1. Introduction

Natriuretic peptides (NPs) reduce cardiac congestion by promoting natriuresis, vasodilation, inhibition of the renin-angiotensin-aldosterone system, and sympathetic nerve activity [1]. Stimulation of myocardial stretch mainly releases NPs into the blood; however, the half-life of NPs is short because the neutral endopeptidase, neprilysin, rapidly degrades NPs. One study using a rat model with heart failure showed increased neprilysin production in the kidneys [2].

Sacubitril valsartan, an angiotensin receptor/neprilysin inhibitor (ARNI), is a recently developed heart failure treatment. It is a fixed-dose combination of a neprilysin inhibitor and an angiotensin II receptor blocker (ARB). ARNI can increase blood NP concentrations as a prodrug of neprilysin inhibitors [3]. Furthermore, ARNI produces an antihypertensive effect. In patients with congestive heart failure, ARNI inhibits atrial remodelling and enhances left ventricular contractility [4,5]. In addition, ARNI improves the symptoms of congestive heart failure associated with these effects and reduces cardiovascular events and mortality from heart failure [6]. In dog heart failure models, ARNI improved left ventricular contractility and reduced myocardial remodelling [7,8], suggesting that ARNI is clinically useful for treating heart failure.

ARNI may benefit the kidneys and heart. In patients with congestive heart failure, ARNI increased the glomerular filtration rate (GFR) [9]. In a double-blinded study of ARNI and enalapril in patients with congestive heart failure and reduced ejection fraction, a lower proportion of patients developed kidney impairment with ARNI than they did with enalapril [10]. A study using mouse models with cardiorenal syndrome also suggested that ARNI reduces renal impairment more than valsartan alone does [11]. However, no studies have investigated the effects of ARNI on kidney function independent of cardiac function in either human or veterinary medicine.

Atrial natriuretic peptide (ANP) augments GFR by dilating glomerular afferent arterioles and constricting efferent arterioles [9]. In contrast, ARBs antagonise the AT1 receptor, are abundant in centrifugal arterioles, dilate centrifugal arterioles, and reduce GFR at the expense of increasing renal blood flow [9]. If neprilysin inhibitors are more effective than ARBs against GFR, ARNI may become a treatment for kidney disease, along with cardiorenal syndrome.

Therefore, we aimed to investigate the effects of ARNI on renal haemodynamics in healthy dogs.

## 2. Results

### 2.1. Creatinine Clearance (CCr)

Table 1 summarises the CCr values before ARNI administration (BL), on Day 7 (D7), and on Day 28 (D28). A significant increase in CCr was observed between BL and D28 and between D7 and D28 (*p* = 0.024 for both) (Figure 1).

### 2.2. C_PAH_

Table 1 lists the C_PAH_ at BL, D7, and D28. No significant difference in C_PAH_ (*p* = 0.097) was observed. Although significance was not achieved (*p* = 0.097), the trend was to have an increase on D28 (Figure 2). In addition, there was little overlap between the BL and D28 95% confidence intervals (Table 1).

### 2.3. Blood Pressure

Table 1 summarises the blood pressure values at BL, D7, and D28. Systolic blood pressure significantly decreased on D28 compared to that on BL (*p* = 0.037, Figure 3).

### 2.4. Plasma ANP Concentration

Table 1 presents the plasma ANP concentrations at BL, D7, and D28. No significant differences in the plasma ANP concentrations (*p* = 0.212) were observed; however, the plasma ANP concentrations of all animals increased on D7. On D28, the plasma ANP concentrations of three of the five animals increased compared to those of the animals on BL. The plasma ANP concentration of one of the remaining two animals remains almost unchanged, whereas that of the other decreased (Figure 4).

### 2.5. Urine Volume 

Table 1 presents the urine output volumes at BL, D7, and D28. Urine output volumes were not significantly different among the BL, D7, and D28 groups (*p* = 0.448, Figure 5).

### 2.6. Heart Rate

Table 1 presents the heart rates at BL, D7, and D28. Heart rate was not significantly different among the BL, D7, and D28 groups (*p* = 0.078; Figure 6).

## 3. Discussion

We investigated the effect of ARNI on renal haemodynamics in healthy dogs and observed that CCr was significantly increased in all dogs on D28 compared to BL. Simultaneously, C_PAH_ increased in all dogs; however, the differences were not significant. Our study, to our knowledge, is the first to determine the effects of ARNI on renal haemodynamics in healthy animal models. GFR and renal plasma flow (RPF) increased after ARNI treatment. This result is similar to those of previous studies using mouse and rat models with type II diabetes [12,13]. Increased glomerular hydrostatic pressure or increased RPF are factors associated with increased GFR. In healthy humans, a myogenic reflex in the afferent arteriole and tubuloglomerular feedback allow PRF and GFR to remain nearly constant, while mean arterial pressure fluctuates between 80 and 160 mmHg [14]. The increase in GFR observed in our study was probably particularly caused by RPF. Sacubitril can increase RPF by dilating afferent arterioles caused by increasing blood levels of NPs such as ANP and valsartan and by dilating efferent arterioles caused by antagonising AT1 receptors [15]. However, in the latter case, the GFR should be reduced. Indeed, monotherapy with ARBs reduces GFR [16]. In contrast, sacubitril can increase the GFR by inhibiting the breakdown of NPs. In a previous study comparing ARNI and valsartan monotherapy groups in patients with heart failure, ARNI significantly increased GFR [9]. Therefore, GFR was associated with increased RPF owing to ARNI treatment in healthy dogs. The increase in GFR observed in our study is probably due specifically to the RPF. Another possible reason for the increase in GFR in our study could be the changes in the imported arterioles due to natriuresis. Previous reports in mice with advanced diabetic kidney disease showed that natriuresis causes increased release of vasoconstrictors and increased release of vasodilators in the imported arteries [17]. Sodium excretion fractions were not sought in the present study. However, given that blood ANP concentrations tended to increase at D28 compared with BL, although not statistically significant, an increase in blood ANP concentrations and consequent increase in natriuresis may have occurred at D28 due to sacubitril. However, although RPF decreases when imported arterioles constrict, RPF increases in this study, so the effect of natriuresis would be slight, if any.

Supporting the hypothesis that the inhibition of NP degradation by sacubitril increases GFR, our study revealed that plasma ANP concentrations increased on D7 compared to BL. This result is similar to those of previous studies investigating the effects of ARNI in humans with heart failure [18]. However, on D28, plasma ANP concentrations increased in three of the five dogs compared to those observed on BL, whereas different plasma ANP concentrations were observed in the remaining two dogs. Individual differences in neprilysin metabolism may explain this observation. Neprilysin, a membrane-bound enzyme, is metabolised by cytochrome P450s (CYPs); however, its activity varies among individuals [19]. Individual differences in the metabolism of neprilysin, a sacubitril target, have been reported [20]. The effect of ARNI on blood NP levels may also be related to the duration of administration, as differences occurred between D7 and D28. Therefore, further studies with longer administration periods are required.

Our study showed that systolic blood pressure decreased significantly on D28 compared to BL. This finding is consistent with previous reports of significant blood pressure reduction following the administration of ARNI to patients with hypertension [21]. In dogs with myxomatous degenerative mitral valve disease, systolic blood pressure decreased after ARNI administration; however, no significant difference was observed [22]. In a study involving dogs with chronic kidney disease (CKD) and proteinuria treated with telmisartan, systolic blood pressure was reduced by a median of −16 mmHg (range: −69–33 mmHg at 120 days from baseline) [23]. Furthermore, our study showed that systolic blood pressure was reduced by a median of −19 mmHg between the BL and D28 measurements, although all dogs were clinically healthy. ARNI may synergise the antihypertensive effect of valsartan with the enhancement of the NP effect of sacubitril and its associated vasodilatory effect. Considering its antihypertensive effect and increased GFR, ARNI may also be helpful in dogs with CKD and decreased GFR. However, the effect of ARNI in dogs with CKD requires further study.

Blood concentrations of NPs increase because of neprilysin inhibition; therefore, they have a diuretic effect. Notably, our study showed no significant difference in urine output after ARNI administration, similar to a report on a diabetic mouse model that showed no significant changes in urine output after ARNI administration [14]. Additionally, patients with heart failure treated with ARNI did not experience weight loss, suggesting that the diuretic effect of ARNI is relatively weak [24]. In dogs, carperitide, an ANP preparation, had a weaker diuretic effect than furosemide [25]. Patients with congestive heart failure may develop kidney impairment because of diuretic-induced dehydration and the associated reduction in renal blood flow [16]. Furosemide is the first choice of diuretics used in veterinary medicine [26]. For dogs whose pulmonary oedema subsides after furosemide administration, transitioning to ARNI is considered, as its diuretic effect is mild and may not reduce GFR.

In this study, heart rate increased on D7 and decreased on D28. The baroreflex effect, owing to the antihypertensive effect, may be a reason the heart rate increased on D7. However, ANP had no significant effect on the heart rate in healthy dogs [27]. Additionally, in a study of telmisartan, an ARB, in healthy dogs, no change in heart rate was observed, although a significant decrease in blood pressure occurred [28]. The sympathoinhibitory effect of ANP on ARNI may explain the decrease in heart rate on D28. However, since a variation in ANP on D28 was observed, the association was considered weak. We believe that further examination of the effects of ARNI on heart rate, including 24 h diurnal fluctuations, is necessary. 

Our study had several limitations. First, the sample size consisted of only five dogs, which might have resulted in limited statistical data. The median clearance at each time may differ as the number of dogs increases; this cannot be entirely ruled out. However, five dogs were considered appropriate, as significant differences were detected in several evaluation parameters, and the study was conducted with careful consideration of animal welfare. Second, the conditions related to drinking water may have influenced specific parameters. Drinking water was not restricted immediately before blood collection to measure plasma ANP concentrations; therefore, variations in the ANP concentrations could occur. Additionally, the lack of uniformity in the amount of drinking water consumed immediately before the start of the clearance study may have affected urine output volume, CCr, and C_PAH_, potentially leading to values that were lower than expected. Nevertheless, the overall impact of these factors was minimal, considering the observed changes in CCr and C_PAH_. Third, serum creatinine and urine creatinine were measured using different methods, the enzymatic method and the Jaffe method, respectively, in this study. Different methods of measuring serum and urine creatinine may have affected clearance results. Finally, we focused exclusively on measuring ANP concentrations; concentrations of other NPs, such as brain natriuretic peptide (BNP), were not assessed. Therefore, the differences in the secretion and metabolism of ANP and BNP may have yielded different results. In addition to comparing ANP and BNP with different modes of secretion, evaluating N-terminal proNPs together may explain changes in cardio-behavioural status. Finally, the dogs in this study were healthy; further studies on ARNI’s hypotensive and renoprotective effects in dogs with glomerulonephritis and hypertension are needed.

## 4. Materials and Methods

### 4.1. Animals

We used five clinically healthy dogs managed in our laboratory of Veterinary Internal Medicine II of Nippon Veterinary and Life Science University (aged 6.46 ± 1.04 years, 11.46 ± 1.64 kg, four unneutered males, and one unneutered female) for this study. Before the experiment, all dogs underwent a physical examination, urinalysis, blood tests (complete blood count and blood chemistry), chest radiography, and echocardiography to determine clinical health. During the study protocol, the dogs were maintained individually in stainless-steel cages, provided ad libitum drinking water, and fed commercial dry food twice daily (at 8:00 a.m. and 5:30 p.m.). We followed the Guidelines for Institutional Laboratory Animal Care and Use of the Nippon Veterinary and Life Science University (approval no. S22-15).

### 4.2. Study Protocol

ARNI (Entresto, Novartis Pharma K.K., Tokyo, Japan) was administered orally to all dogs at a dose of 20 mg/kg twice daily for 28 days [7]. A series of tests and examinations were conducted the day before BL, on D7, and on D28. We monitored exogenous CCr, C_PAH_, blood pressure, and plasma ANP concentration during the three periods. CCr and C_PAH_ were performed for three consecutive sets of 20 min per monitor (including urine volume monitoring). We also conducted physical examinations (e.g., body weight, heart rate, respiratory rate, and dehydration assessment). Samples were preserved according to the institution’s guidelines. Plasma ANP concentrations fluctuate diurnally in humans; therefore, 10:00 a.m., a less variable time point, was set as the time for blood sampling [29]. Figure 7 illustrates the detailed experimental procedure.

### 4.3. CCr and C_PAH_

Exogenous CCr testing is a valuable assessment tool for GFR in dogs [30]. Creatinine (Cre) was aseptically dissolved in 0.9% sterile saline at a final concentration of 25 mg/mL. Subsequently, Cre was administered as a bolus dose of 60 mg/kg over 2 min via the radial skin vein, followed by continuous infusion of Cre solution (3.2 mg/mL) at a dose of 16 mg/kg/h. Complete urination occurred after a 45-min equilibration period from the start of administration. Urine volumes were recorded at 20 min, and a portion of the urine was kept as a sample to determine the Cre concentration. This study determined urine volumes by averaging 20 min data obtained during three clearances. The bladder was emptied at designated time points, and the same individual collected all urine from each dog to quantify the exact urine volume. Following removal of urine, the bladder was rinsed with 5 mL sterile saline, and the rinse solution was collected. Blood sampling to measure plasma Cre concentrations was performed 10 min after the complete measurement of excretion. Blood samples were collected in heparin tubes (Fujifilm Corporation, Tokyo, Japan), and the plasma was centrifuged at 3000 rpm for 5 min. The urine supernatant sample was collected after centrifugation at 1500 rpm for 5 min. The storage temperature of the plasma and urine supernatants was −80 °C until the measurement of Cre concentrations. Urinary Cre concentrations were measured using Lab Assay™ Creatinine (Fujifilm Wako Pure Chemicals, Osaka, Japan). Plasma Cre concentrations were determined using a Fuji Drychem 4000 V (Fujifilm Medical Corporation, Tokyo, Japan). 

The C_PAH_ is a renal plasma flow (RPF) evaluation tool. The drug used to determine clearance was a para-amino sodium hippurate injection (para-amino sodium hippurate injection 10%, Daiichi Sankyo Pharmaceutical Co., Ltd., Tokyo, Japan). Sodium para-amino equine urate (10 mg/mL) was diluted 10-fold in 0.9% sterile saline and delivered at a rate of 6.0 mg/kg via the radial cutaneous vein over 5 min, followed by a PAH solution (10 mg/mL) at a rate of 2 mL/min. Urine and blood samples were collected simultaneously by using an exogenous CCr test.

The PAH concentrations were determined according to a previous study [31]. Sodium para-amino equine urate solutions (prepared blanks) adjusted to concentrations of 0.1, 0.3, 1.0, 3.0, and 10 µg/mL were dispensed into six microtubes each. A mixture of 100 µL of sodium para-amino equine uric acid solution and 700 µL of ion-exchanged water (800 µL blank) was used to adjust the concentrations of sodium para-amino equine urate solutions. Next, 200 µL of 15 g/dL trichloroacetic acid was added dropwise to all six well-mixed microtubes. The mixture was allowed to stand for 10 min and then centrifuged at 3000 rpm for 5 min. The microplate wells were loaded with 250 µL of each of these supernatants. Subsequently, 50 µL of 2N HCl and 7.5 µL of 0.2 g/dL sodium nitrite solution were added dropwise, mixed, and allowed to stand for 3 min. Excess sodium nitrite was removed by adding 15 µL of 25 g/dL urea solution, stirring well, and allowing the solution to stand for 10 min. Finally, 10 µL of Tsuda’s reagent was introduced and allowed to stand for 5–10 min, and the absorbance was measured at 570 nm to establish a calibration curve using POWERSCAN®︎HT (Dainippon Pharmaceutical Co., ltd, Osaka, Japan). After the calibration curves were prepared, the actual concentrations in the plasma and urine samples were measured. Plasma supernatant obtained after centrifugation and urine, which was diluted 1000-fold, were dispensed in 250 µL quantities into the wells of a microplate. The absorbance was measured using the wavelength that generated the calibration curve, and the plasma PAH concentration was calculated from the calibration curve.

CCr and C_PAH_ were calculated using the following equations:$$\mathrm{CCr}\;(\mathrm{mL}/\min)=\frac{Urine\;volume\left(mL\right)\times Urinary\;Cre\;concentration\;(mg/dL)}{Plasma\;Cre\;concentration\left(mg/dL\right)\times Body\;weight\left(kg\right)\times Urine\;storage\;time\;(20\;min)}$$
$$C_{\mathrm{PAH}}\;(\mathrm{mL}/\min/\mathrm{kg})=\frac{Urine\;volume\left(mL\right)\times Urinary\;PAH\;concentration\;(mg/dL)}{Plasma\;PAH\;concentration\left(mg/dL\right)\times Body\;weight\left(kg\right)\times Urine\;storage\;time\;(20\;min)}$$

### 4.4. Blood Pressure

Blood pressure was measured using an oscillometric sphygmomanometer (Veterinary Biometric Monitor BP-608v; Fukuda ME Co., Ltd., Tokyo, Japan). Three consecutive consistent readings were recorded, and the average of these measurements was calculated.

### 4.5. Plasma ANP Concentration

Blood samples (2 mL) were aliquoted into tubes containing EDTA-2Na + aprotinin and centrifuged at 3000 rpm for 5 min. The plasma sample was stored at a temperature of −80 °C until measurement. An external institute measured the plasma ANP concentration (Fujifilm VET Systems Co., Ltd., Tokyo, Japan).

### 4.6. Statistical Analysis

Statistical analyses were performed using commercially available software (SPSS Statistics 24, IBM Japan, Tokyo; EZR, Saitama Medical Centre, Jichi Medical University, Saitama, Japan). The Shapiro–Wilk test was used to determine the normality of each variable. One-way ANOVA or the Friedman test was used to compare each variable, with post hoc correction using the Steel–Dwass test, where appropriate. Parameters with significant differences were presented as a median range, whereas other parameters were presented as a median interquartile range.

Statistical significance was considered at *p* < 0.05.

## 5. Conclusions

ARNI can affect renal haemodynamics in healthy dogs, as increased GFR and RPF were observed. Future studies are needed to assess the effects of ARNI in dogs with glomerulonephritis or hypertension or in dogs with chronic kidney disease with concomitant heart disease.

## Figures and Tables

**Figure 1 ijms-25-06169-f001:**
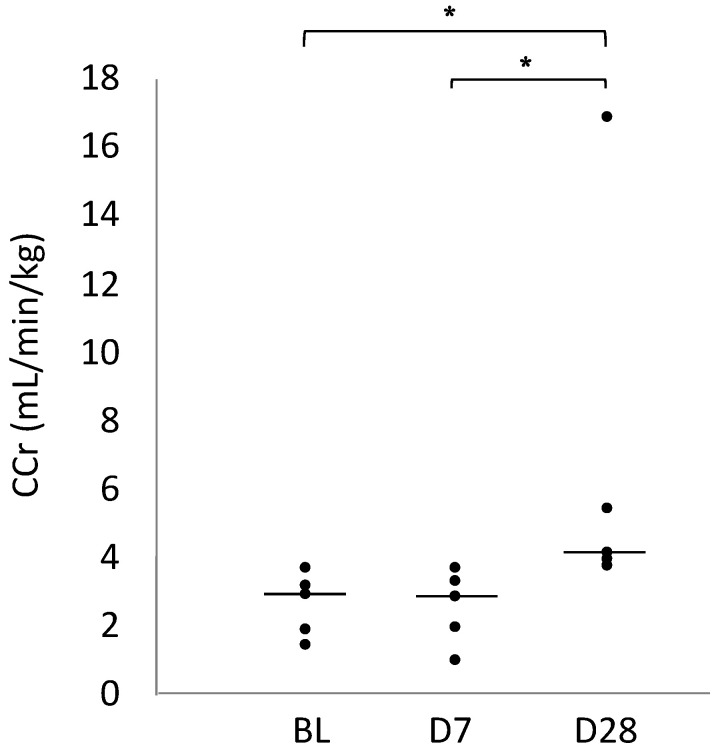
Scatter diagram comparing the CCr for BL, Day 7, and Day 28. In the plot, the centre line represents the median. * *p* < 0.05.

**Figure 2 ijms-25-06169-f002:**
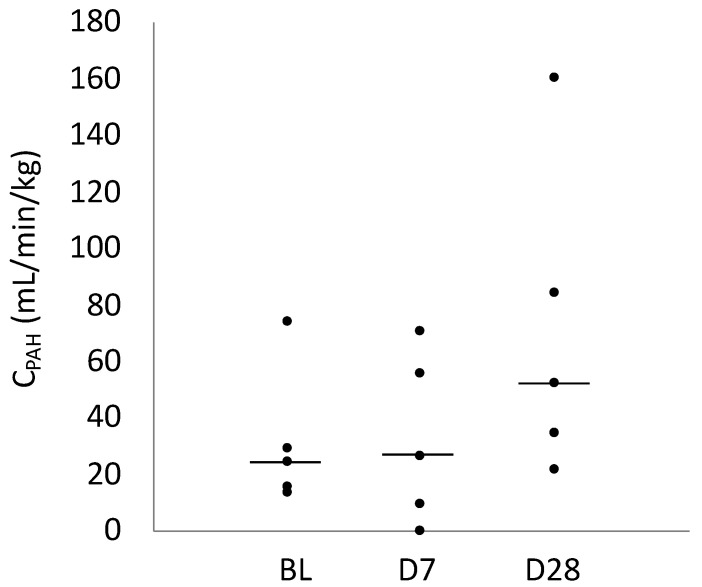
Scatter diagram comparing the C_PAH_ on BL, Day 7, and Day 28. In the plot, the centre line represents the median.

**Figure 3 ijms-25-06169-f003:**
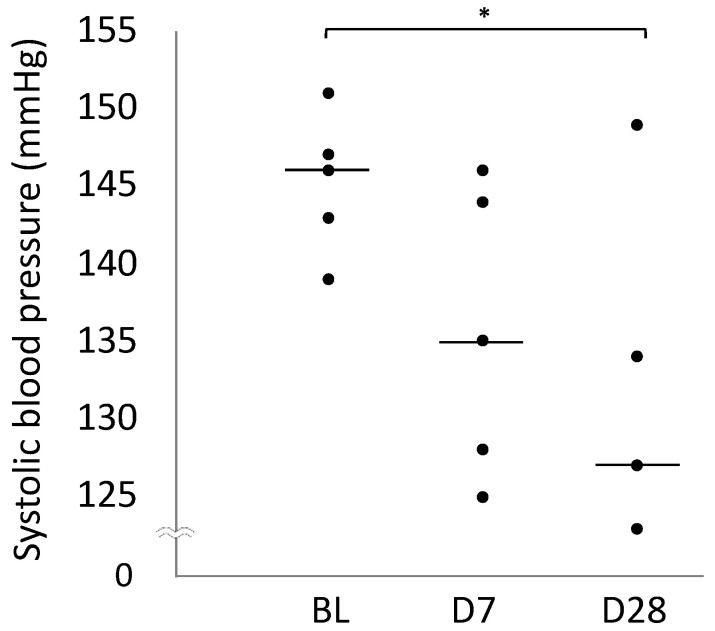
Scatter diagram comparing systolic blood pressure on BL, Day 7, and Day 28. In the plot, the centre line represents the median. * *p* < 0.05.

**Figure 4 ijms-25-06169-f004:**
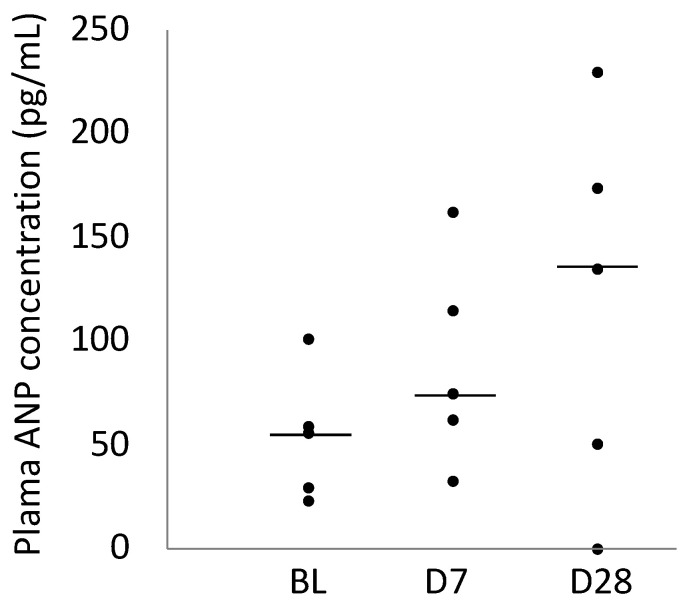
Scatter diagram comparing plasma ANP concentrations on BL, Day 7, and Day 28. In the plot, the centre line represents the median.

**Figure 5 ijms-25-06169-f005:**
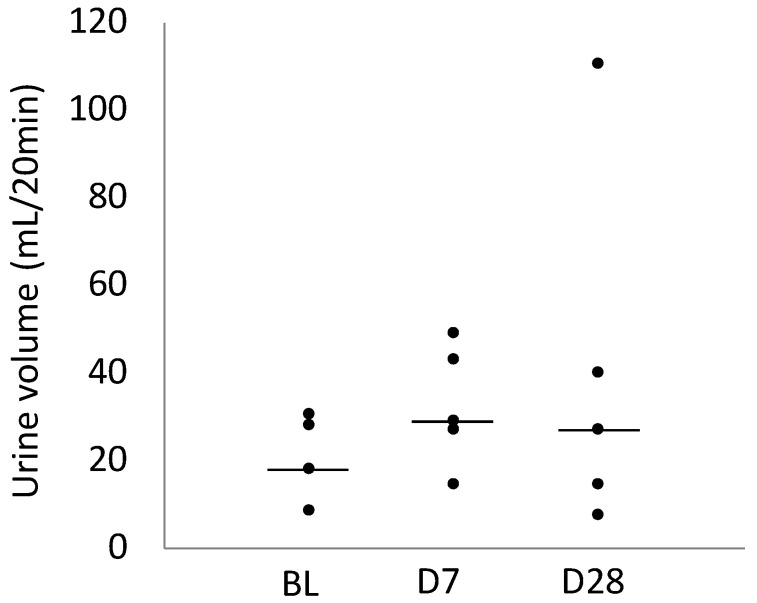
Scatter diagram comparing urine output on BL, Day 7, and Day 28. In the plot, the centre line shows the median.

**Figure 6 ijms-25-06169-f006:**
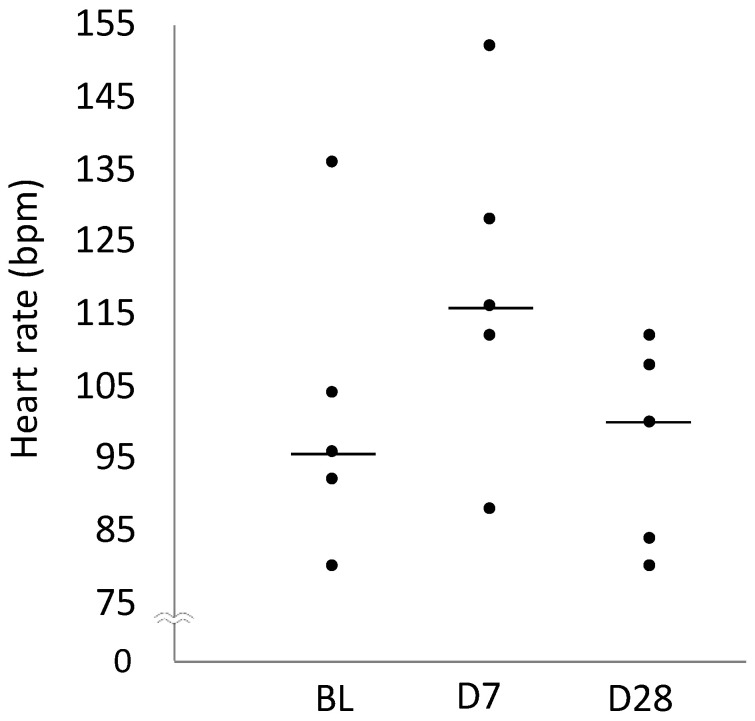
Scatter diagram comparing heart rate on BL, Day 7, and Day 28. In the plot, the centre line shows the median.

**Figure 7 ijms-25-06169-f007:**
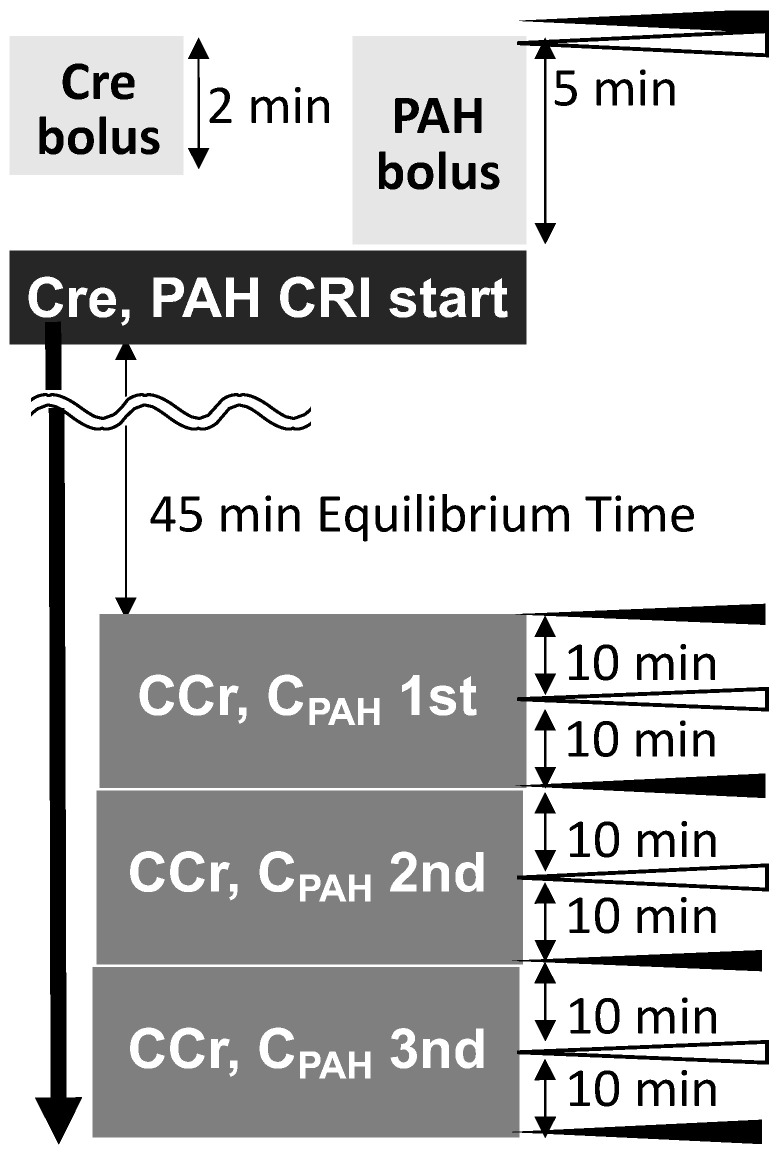
Experimental procedure. We performed this process on the day before administration (BL), Day 7, and Day 28. CRI, continuous rate infusions; black arrowhead, the timing of complete urination and urine sampling; white arrowhead, the timing of blood sampling.

**Table 1 ijms-25-06169-t001:** Comparison of CCr, C_PAH_, heart rate, blood pressure, plasma ANP concentration, and urine output volume at BL, Day 7, and Day 28.

	BL	D7	D28
CCr (mL/min/kg)	2.88 (1.87–3.19)	2.81 (1.95–3.32)	4.14 (3.91–5.42) *
95% Confidence interval	1.77–3.42	1.60–3.51	3.53–5.07
C_PAH_ (mL/min/kg)	24.92 (16.02–29.61)	26.43 (9.81–55.74)	52.49 (34.87–84.48)
95% Confidence interval	13.55–28.53	6.15–58.98	21.0–75.73
Systolic blood pressure (mmHg)	146 (143–147)	135 (128–144)	127 (127–134) *
95% Confidence interval	141–149	127–143	123–142
Mean blood pressure (mmHg)	98 (98–102)	94 (93–105)	97 (94–98)
95% Confidence interval	97–101	89–105	91–101
Diastolic blood pressure (mmHg)	76 (75–79)	78 (67–86)	80 (78–81)
95% Confidence interval	73–78	67–86	82–73
Plasma ANP concentration (pg/mL)	56 (30–59)	74 (62–115)	135 (50–173)
95% Confidence interval	26–80	44–133	36–198
Urine Volume (mL/20 min)	18.4 (18.1–28.1)	29.4 (27.6–43.5)	27.2 (14.9–111.0)
95% Confidence interval	13.2–28.4	20.9–44.9	3.8–76.5
Heart rate (bpm)	96 (92–104)	116 (112–128)	100 (84–108)
95% Confidence interval	83–120	98–139	84–109

ANP, atrial natriuretic peptide; CCr, creatinine clearance; BL, before ARNI administration; C_PAH,_ para-amino hippuric acid-clearance. Data indicated as median (interquartile range). 95% confidence intervals are shown as min–max. * *p* < 0.05.

## Data Availability

The data presented in this study are available upon request from the corresponding author.

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
