# Peer review of "Study on the Effects of Angiotensin Receptor/Neprilysin Inhibitors on Renal Haemodynamics in Healthy Dogs"

_ijms, 2024, doi:10.3390/ijms25116169_

Round 1

Reviewer 1 Report

Comments and Suggestions for Authors

This reviewer would like to thank the authors for a project that was well developed and presented in a clear and concise manner. This pilot study should definitely prompt additional studies with a larger cohort of normal dogs as well as studies in dogs with both chronic kidney disease as well as those with protein-losing nephropathies.  

Ther are some areas within the manuscript that should be further clarified and address by the authors to facilitate better understanding by readers.

In Abstract and Results

Line19 and Lines 66-69: You state that GFR increased significantly in all dogs on BL,D7 and D28. But how can GFR increase before administration of the drug.  You state that hemodynamic studies were performed the day before ARNI administration.  So what you are actually reporting is the "normal" or "starting"  GFR for the study dogs.  The median of those dogs GFR  based on CCr was 2.88 ml/kg/day. On D7 the median was 2.81. You report that all dogs had an increase in CCr. Actually it would appear based on both Table 1 and  Figure 1 that there was no significant difference between BL and D7 CCr ( GFR).  This reviewer does agree that there is a significant increase between BL and D28 and between D7 and D28. This wording and interpretation should be clarified.

Abstract Line 22:  Improved wording would be "D7 compared to BL" rather than " D7 than on BL".

Results Line 68-69:  Rewording of sentence should be "  there was an overall increase in CCr between BL and D28 , with a greater increase seen between D7 to D28.  Or are you stating that the delts increase from D7 to D28 was greater than when comparing the delta increase in BL to Day 7 or BL to day 28?  Please clarify.

Results section CPAH Lines 105-106:  Sentence or rather the thought is not complete. As written no significant difference was observed however , the CPAH increased on D7 and D28  compared to that on BL.   
This means that you have a one sentence paragraph which should always be avoided.  You might add that the although significance was not achieved P=0.097  that the trend was to have an increase on D28.  Eyeing the figure it looks like there maybe a significance between BL and D28.  Are you sure that your statistical calculation is correct?

The discussion was well written. The only area that this reviewer would like to perhaps see expansion of the discussion is in the first paragraph discussion on GFR. Lines 242-242: The authors comment on the increase in GFR observed in your study was probably particularly caused by RPF. There should be a bit more support to this supposition. Might you  offer other possible explanations or rule or more  to bolster support for your hypothesis? Could the small numbers of dogs have actually affected outcome. Between the testing periods there is quite a bit of overlap- more dogs in the study may change the median CCr.  Was the same method utilized for measuring serum creatinine vs urine creatinine? Jaffe method or other?  Was same individual collecting all urine from the dogs to ensure total bladder emptying and exact urine output quantification at the designated time points?

Methods Section:  

Lines 412-415:   In both formulas for CCr and CPAH - please clarify the urine storage time ( min).   I think that you are referring to 10 min between collection and not the time that the urine was stored in a freezer prior to  analysis.    Typical clearance studies utilize the collection period time frame to refer to the volume of  urine produced  over x number of minutes .

Comments on the Quality of English Language

The quality of the English language in the manuscript is fine except for the few instances listed above and that is not use of the language per say but just confusion with wording of the sentence structure.

Author Response

Manuscript ID: ijms-2985486

Article Title: Study on the Effects of Angiotensin Receptor-Neprilysin Inhibitors on Renal Haemodynamics in Healthy Dogs

Responses to Reviewer 1:

We are grateful for your comments and useful suggestions that have helped us in improving our manuscript. Following suggestions from reviewers, the content has been revised. The changes to manuscript within the document are highlighted in red coloured text. The following changes have been made in response to points raised by reviewer 1;

In Abstract and Results Line19 and Lines 66-69: You state that GFR increased significantly in all dogs on BL, D7 and D28. But how can GFR increase before administration of the drug.  You state that hemodynamic studies were performed the day before ARNI administration.  So what you are actually reporting is the "normal" or "starting" GFR for the study dogs.  The median of those dogs GFR based on CCr was 2.88 ml/kg/day. On D7 the median was 2.81. You report that all dogs had an increase in CCr. Actually it would appear based on both Table 1 and Figure 1 that there was no significant difference between BL and D7 CCr (GFR).  This reviewer does agree that there is a significant increase between BL and D28 and between D7 and D28. This wording and interpretation should be clarified.

Response to Line19 and Lines 66-69: We sincerely apologize for the mistake. As you point out, although the summary stated,' GFR increased significantly in all dogs on BL, D7 and D28, ' the only fundamental significant differences were between BL and D28 and between D7 and D28. We have, therefore, altered the documentation in the summary (Line 19-20).

Abstract Line 22:  Improved wording would be "D7 compared to BL" rather than " D7 than on BL".

Response to Line 22: We appreciate your advice and have rephrased it as a suggestion (Line 22).

Results Line 68-69:  Rewording of sentence should be " there was an overall increase in CCr between BL and D28 , with a greater increase seen between D7 to D28.  Or are you stating that the delts increase from D7 to D28 was greater than when comparing the delta increase in BL to Day 7 or BL to day 28?  Please clarify.

Response to Line 68-69: As you point out, this sentence was very vague and difficult to convey to the reader. In fact, we only wanted to convey that both D7 and D28 showed higher CCr values than BL. The sentence has therefore been deleted. I apologize for the confusion (Line 68).

Results section CPAH Lines 105-106:  Sentence or rather the thought is not complete. As written no significant difference was observed however , the CPAH increased on D7 and D28  compared to that on BL.   
This means that you have a one sentence paragraph which should always be avoided.  You might add that the although significance was not achieved P=0.097  that the trend was to have an increase on D28.  Eyeing the figure it looks like there maybe a significance between BL and D28.  Are you sure that your statistical calculation is correct?

Response to Line 105-106: We thank you for your helpful advice. Indeed, the statistical evidence that PAH increased in D7 and D28 compared to BL was poor. Therefore, as you advised, we have changed the statement to "Although significance was not achieved P=0.097  that the trend was to have an increase on D28". We added that to the text because the confidence intervals for BL and D28 showed little overlap. Against this background, confidence intervals have been added for all parameters in the revised manuscript (Line 73, 99-101, Table 1).

Discussion section Lines 242-242: 1) The authors comment on the increase in GFR observed in your study was probably particularly caused by RPF. There should be a bit more support to this supposition. Might you offer other possible explanations or rule or more to bolster support for your hypothesis? 2) Could the small numbers of dogs have actually affected outcome. Between the testing periods there is quite a bit of overlap- more dogs in the study may change the median CCr. 3) Was the same method utilized for measuring serum creatinine vs urine creatinine? Jaffe method or other?  4) Was same individual collecting all urine from the dogs to ensure total bladder emptying and exact urine output quantification at the designated time points?

Response to Line 242-242:

1) To further our discussion on increased GFR, we have added a detailed description of the hypothesis and other possible explanations. Specifically, we added additional references and text to support the statement, "In normal kidneys, autoregulatory mechanisms of renal blood flow are thought to maintain a constant GFR despite fluctuations in blood pressure". Secondly, another consideration of natriuresis by sacubitril and the associated increase in GFR was added. However, in this case, we summarised that the increase in GFR is likely related to the increase in PRF, as PRF is reduced (Line 236-238, 246-256).

2) As you point out, we cannot completely rule out the possibility that the CCr results may change due to increased dogs incorporated. Therefore, this has been added to the limitation (Line 305-306).

3) In this study, serum creatinine and urine creatinine were measured using different methods: the enzymatic method and the Jaffe method, respectively. We did not mention the possibility that different measurement methods could influence the results, so we added to the limitation that different measurement methods were used for serum creatinine and urine creatinine (Line 316-319).

4) In this study, the bladder was emptied at designated time points, and the same individual collected all urine from the dog to quantify the exact urine volume. We added this to the article (Line 386-388). To accurately record urine volume, we removed the urine in the bladder with a catheter, after which the bladder was further flushed with 5 mL of sterile saline, and the flushing solution was also collected (Line 388-389).

Methods Section:  Lines 412-415:   In both formulas for CCr and CPAH - please clarify the urine storage time (min).   I think that you are referring to 10 min between collection and not the time that the urine was stored in a freezer prior to analysis. Typical clearance studies utilize the collection period time frame to refer to the volume of urine produced  over x number of minutes .

Response to Line 412-415: Although the description in the text states Line 386 that "This study determined urine volumes by averaging 20-minute data obtained during three clearances.", both the CCr and CPAH equations specify the urine storage time (minutes) was not specified. The urine storage time in this study is 20 minutes; this has been added to both the CCr and CPAH equations (Line 425-428).

In addition to the above, we have revised the article in response to reviewer 1. We added the reference, providing evidence that exogenous creatinine lance clearing benefits dogs (Line 379, 548-550).

Thank you very much for your helpful comments.

Sincerely,

Reviewer 2 Report

Comments and Suggestions for Authors

I my opinion the manuscript “Study on the effects of angiotensin receptor-neprilysin inhibitors on renal haemodynamics in healthy dogs” fulfills the requirements set for publication inInternational Journal of Molecular Sciences” and I recommend publishing it in the present form.

Presented study aimed to investigate the effect of angiotensin receptor neprilysin (ARNI) on renal haemodynamics in 5 healthy dogs. The authors found out that after administration of ARNI glomerular filtration rate (GFR) significantly increased, renal plasma flow (RPF) and plasma atrial natriuretic peptide (ANP) concentration increased, systolic pressure significantly decreased and the urine output volume and heart rate remained relative stable. Based on the obtained results the author conclude that ARNI may enhance renal haemodynamics in healthy dogs. The authors suggest that in the future ARNI could be a valuable drug for treating heart and kidney diseases  in dogs. The authors themselves point out that the study had a several limitation. Nevertheless the obtained result are very promising and there is the need for the further study in this area.

In the experiment the appropriate methods were used, the results and discussion were clearly and openly described, the planned goals were achieved.

Actually, I was wondering why to determine the GFR the authors do not used exogenous inulin instead of exogenous creatinine?

Best regards,

Author Response

Manuscript ID: ijms-2985486

Article Title: Study on the Effects of Angiotensin Receptor-Neprilysin Inhibitors on Renal Haemodynamics in Healthy Dogs

Responses to Reviewer 2:

We are grateful for your comments and the useful suggestion that have helped us in improving our manuscript. The changes to manuscript within the document are highlighted in red coloured text. Following suggestions from reviewers, the content has been revised. The following changes have been made in response to points raised by reviewer 2;

Actually, I was wondering why to determine the GFR the authors do not used exogenous inulin instead of exogenous creatinine?

Response

A previous study suggests that inulin clearance may not be adequate in dogs. On the other hand, in this study, the average recovery of exogenous creatinine in canine urine was approximately 99%, similar to another report, indicating that extrarenal excretion of creatinine is negligible. Based on these previous reports, we chose exogenous creatinine clearance rather than inulin clearance.

As you point out, we did not state the literature on which we based our choice of exogenous creatinine clearance. We have, therefore, added the supporting literature to the article (Line 379, 548-550).

In addition to the above, we have revised the article in response to reviewer 1.

The main changes are

  • A textual modification (Line 19-20, 22, 68, 305-306, 316-319, 386-388, 425-428)
  • The addition of 95% confidence intervals for each result (Table 1, Line 73, 91-101)
  • The first paragraph of the discussion should be added (Line 236-238, 246-256)

Thank you very much for your helpful comments.

Sincerely,